# Influence of Nanoparticles on the Dielectric Response of a Single Component Resin Based on Polyesterimide

**DOI:** 10.3390/polym14112202

**Published:** 2022-05-28

**Authors:** Štefan Hardoň, Jozef Kúdelčík, Anton Baran, Ondrej Michal, Pavel Trnka, Jaroslav Hornak

**Affiliations:** 1Department of Physics, Faculty of Electrical Engineering and Information Technology, University of Žilina, 010 26 Žilina, Slovakia; jozef.kudelcik@feit.uniza.sk; 2Department of Physics, Faculty of Electrical Engineering and Informatics, Technical University of Košice, Park Komenského 2, 042 00 Košice, Slovakia; anton.baran@tuke.sk; 3Department of Materials and Technology, Faculty of Electrical Engineering, University of West Bohemia, 306 14 Pilsen, Czech Republic; mionge@fel.zcu.cz (O.M.); pavel@fel.zcu.cz (P.T.); jhornak@fel.zcu.cz (J.H.)

**Keywords:** polyesterimide, zinc oxide, aluminium oxide, dielectric spectroscopy, dielectric relaxation, ^1^H NMR measurements

## Abstract

The influence of various types of nanoparticle fillers with the same diameter of 20 nm were separately incorporated into a single component impregnating resin based on a polyesterimide (PEI) matrix and its subsequent changes in complex relative permittivity were studied. In this paper, nanoparticles of Al${}_{2}$O${}_{3}$ and ZnO were dispersed into PEI (with 0.5 and 1 wt.%) to prepare nanocomposite polymer. Dielectric frequency spectroscopy was used to measure the dependence of the real and imaginary parts of complex relative permittivity within the frequency range of 1 mHz to 1 MHz at a temperature range from +20 °C to +120 °C. The presence of weight concentration of nanoparticles in the PEI resin has an impact on the segmental dynamics of the polymer chain and changed the charge distribution in the given system. The changes detected in the ${}^{1}$H NMR spectra confirm that dispersed nanoparticles in PEI lead to the formation of loose structures, which results in higher polymer chain mobility. A shift of the local relaxation peaks, corresponding to the α-relaxation process, and higher mobility of the polymer chains in the spectra of imaginary permittivity of the investigated nanocomposites was observed.

## 1. Introduction

Polymeric materials have directly or indirectly become an integral part of our lives in recent years, as they are widely used in various areas of our daily life. A group of materials that consist of a polymer matrix and intentionally dispersed fillers of nanometer size (up to 100 nm) are hybrid systems that are defined as polymer nanocomposites (NCs).The efforts of many scientists in recent years have also been aimed at achieving a synthesis of different types of polymer nanocomposites and understanding their basic principles in determining their properties for application in a wide range of fields [1,2,3,4,5]. Many large-scale scientific studies carried out in recent years clearly show that when the nanofillers are uniformly dispersed in a polymer matrix, the NCs exhibit remarkably better electrical, thermal [6,7], mechanical [8,9,10] properties compared with a pure polymeric organic matrix or micro composites [11,12,13]. A number of scientific studies have been conducted showing an improvement in the dielectric parameters of polymers (polyethylene, epoxy resins, polyimide, polyamide and PEI) used as electrical insulators when particles of nanometric size have been dispersed into them. The polymer matrix is most often doped with inorganic nanoparticles, which are usually metal oxides, such as SiO${}_{2}$, MgO, ZnO, Al${}_{2}$O${}_{3}$ and TiO${}_{2}$ or nitrides such as BN [3,14,15,16,17,18]. Thus, the type of nanoparticles incorporated, not only the polymer matrix itself, also has an influence on the resulting dielectric properties of the investigated NCs. Research and development of new polymer nanocomposite materials creates new opportunities in nanotechnology suitable for industrial [19] and medical applications [20].

PEI resins are thermoset polymers used in a wide range of mechanical, construction and electrical applications. The continuous development of resin materials and the improvement of their technical properties, including in the field of dielectric parameters, plays an important role in the field of electrical insulation [21,22]. The condition of the insulation system is one of the main indicators of the operational reliability of power electrical equipment in the power industry. The lifetime of rotating and non-rotating machines also depends on the condition of their insulation system, as these systems continuously withstand thermal, mechanical and electrical stresses during their operation [22,23]. Studies aimed at tracking changes in the complex permittivity of NCs as a function of temperature, electric field magnitude and its frequency are fundamental to the characterization of dielectric systems [24,25].

A number of different processes (such as the α-, β- and γ-relaxations and the Maxwell-Wagner-Sillars (MWS) effect) in an AC electric field, the intermediate dipolar effect (IDE), and DC conductivity were described in [26] relation to polarisation mechanisms. It is generally observed that molecule-to-molecule motion among chains is activated during the glass-transition process, and that the α-relaxation observed on NCs is connected with the glass transition process. As the polymer sample transitions from a glassy state to a highly elastic state, the long chains are unfrozen and start to move directionally under AC electric fields [13].

Attempts to modify polyesterimide resin with nanoparticles with focus on the changes in their dielectric properties have been superficially described in the literature, especially when nanosilica [3,27], and occasionally other oxides such as ZnO, TiO${}_{2}$, or Al${}_{2}$O${}_{3}$[28,29] were incorporated to the neat resin. However, the main aim of this study is to provide new knowledge on the behavior of synthesized polymer NC materials based on a commercially used PEI with dispersed Al${}_{2}$O${}_{3}$ and ZnO nanoparticles at the various frequencies and temperatures accompanied by other results such as from dynamic mechanical thermal analysis (DMA) or proton nuclear magnetic resonance (NMR).

## 2. Materials and Methods

As reported in the works [30,31], on the preparation of composite materials based on PEI matrix modified with nanofillers, there are two basic mixing mechanisms. The first mentioned is simple mixing method, when random distribution of the particles of the initial components occurs without changing their physical dimensions in a mixture with random particles. Second mechanism of process is dispersive mixing is used to decrease aggregates of solid particles within the liquid polymer and distribute them within its volume.

A series of technological laboratory experiments were conducted to prepare the NC materials. The direct dispersion method was used for this purpose. As was demonstrated in previous studies [26,32,33], this mechanical method, coupled with other technological procedures, can lead to proper dispersion of nanoparticles in polymer matrices. A controlled dispersion procedure using ultrasound was used next as an efficient, clean and rapid technological procedure. The synthesis of various NCs using this method combinated with ultrasound is also widely used to achieve uniform dispersion in other presented articles [25,34,35,36,37,38,39,40,41,42,43]. They are also preferred in economically sensitive areas, as they are not expensive in terms of experimental equipment.

### 2.1. Base Materials and Sample Preparation

For sample preparation, a single component impregnating resin (the UP 343 from Elantas a member of ALTANA [44]) based on PEI was used. This type of single component impregnating resin is stable and low emission, has crack free curing in thick layers and is applicable up to thermal load according to the thermal class 200 (IEC 60085). Selected parameters from the material data sheet of the used PEI resin are shown in Table 1.

Due to its physico-chemical properties, the selected PEI is mainly used for filling preheated objects (40–50 °C) of all conventional rotating and stationary windings (cavities of all kinds and the construction of electrical equipment). The final investigated NCs consist of a single component PEI resin with dispersed 0.5 wt.%, and 1 wt.% nanoparticles of: (i) zinc oxide (ZnO) [45], and (ii) aluminum oxide (Al${}_{2}$O${}_{3}$) [46] with the same diameter (20 nm). The basic parameters of used nanoparticles are listed in Table 2.

The nanofillers used in this experiment were also selected on the base of previous works [11,13,26,32,47,48], where they provide improvement of the insulating and dielectric properties of the studied NCs. Due to their experimental results, the NCs with these nanoparticles are adapted to the strict insulation requirements of high-voltage power equipment to meet the demands of growing power generation and consumption.

Process of the preparation of PEI NCs is showed in Figure 1. Prior to actual mechanical mixing of the nanoparticles with PEI, the individual nanoparticles were dried in a laboratory vacuum hot air dryer for 24 h to lose their surface moisture. To obtain better viscosity during mechanical mixing, the PEI was heated to 50 °C. In order to prepare the required weight concentrations of the nanocomposite samples (0.5 and 1.0 wt.%), it was necessary to determine the exact weight of the individual dried nanoparticles that were added to the PEI matrix. The PEI matrix base and individual nanoparticles were mixed mechanically for 5 h at 50 °C using a magnetic stirrer (300 rpm); additional mechanical mixing associated with the use of an ultrasonic needle (1 h) and then a vacuuming process for 3 h (10 mbar, 300 rpm) followed. To ensure further controlled and uniform dispersion in the prepared suspension, an ultrasonic needle (20 kHz) was applied for 1 h simultaneously with a magnetic stirrer (300 rpm). The task of the vacuuming process is to remove all air bubbles in the prepared suspension. The use of this method guarantees the dispersion of the nanofillers in the PEI matrix even at very low nanoparticle concentrations. The finished mixture suspension was poured into pre-heated flat circular shaped molds. Then the samples were cured in hot air oven for more than 1 h at 150 °C. Each set of prepared samples contained five samples with a specific nanoparticle concentration for replication and confirmation of the measured results. The process of preparation of the PEI NCs is shown in Figure 1a. The uniform dispersion of particles, or agglomerates thereof, is shown in Figure 1b, where the SEM pictures are shown after the additional adjustment of the color scale, brightness, contrast and gamma correction in the graphics software to highlight ZnO and Al${}_{2}$O${}_{3}$ particles when the polymer matrix is ideally displayed as a black area.

### 2.2. NMR Measurements

The broad line (BL) ${}^{1}$H NMR measurements were carried out on a Varian 400 MHz solid-state NMR spectrometer (Palo Alto, CA, USA) using a probe head with 4 mm ZrO${}_{2}$ rotors at ambient temperature 23 °C and 90 °C. The chemical shifts in all spectra were referenced to tetramethylsilane using adamantane as the external standard. The BL ${}^{1}$H NMR spectra were recorded at ${}^{1}$H resonance frequency of 400 MHz, a π/2 pulse with 2.9 μs duration, 6 s recycle delay and 20 ms acquisition time were applied.

### 2.3. DMA Measurements

For DMA measurements, a DMA Q800 (TA Instruments, New Castle, DE, USA) analyzer was used in dual cantilever operational mode at an amplitude of 5 μm and a frequency of 1 Hz. The temperature range was fixed from 30 to 140 °C with a heating rate of 3 °C/min.

### 2.4. Dielectric Response Measurement

The frequency-temperature dependencies of the real and imaginary parts of the complex relative permittivity were measured within the frequency range of 1 mHz to 1 MHz with increasing temperature from +20 °C to +120 °C. IDAX 350 dielectric analyzer (Group Limited, Dallas, TX, USA) and a QuadTech 7600+ precision LCR meter (IET Labs, Inc., Boston, MA, USA) with a self-designed three-electrode system were used for this purpose, like in previous works [26,32].

## 3. Results

### 3.1. NMR Spectral Analyses of PEI Composites

The BL ${}^{1}$H NMR spectra of solid-state polymers are broadened mainly due to the strong intra- and intermolecular ${}^{1}$H-${}^{1}$H dipolar interactions between neighboring nuclei. Therefore, the width of the ${}^{1}$H NMR signals reflects the intensity of the dipolar interactions in the proton spin system. The BL ${}^{1}$H NMR spectrum of polymers measured below the glass-transition temperature ($T_{g}$) generally consists of one broad line, and above the $T_{g}$ the spectrum is usually the superposition of one broad and one or more narrow lines. The broad line corresponds to the hydrogen nuclei in the rigid polymer chains, while the narrow lines can be associated with the hydrogen nuclei located in the mobile chains, in which intense motion averages the dipolar interactions.

The BL ${}^{1}$H NMR spectra for pure PEI and its mixtures with Al${}_{2}$O${}_{3}$ or ZnO nanoparticles measured at ambient temperature and at 90 °C are depicted in Figure 2. At ambient temperature, a single unresolved broad resonance is observed for all studied samples and only negligible changes were visible in the shape of the BL ${}^{1}$H NMR spectra comparing pure PEI and its mixtures. Broad resonances originate from the strong ${}^{1}$H-${}^{1}$H dipolar interactions present in these rigid polymers. This can be expected due to the fact that the ambient temperature is far below the $T_{g}$ of the studied samples. The $T_{g}$ values for all studied PEI and its nanocomposites were estimated from the peak in temperature dependence of tan δ (Figure 3) and their values were in the range 93–113 °C.

On the other hand, the BL ${}^{1}$H NMR spectra of PEI and its mixtures measured at 90 °C have a distinctly different shape than the spectra measured at ambient temperature. Generally, at a temperature close to $T_{g}$, the magnitude and rate of the polymer chains fluctuations rapidly increase, leading to the motional averaging of the dipolar interactions and the narrowing of broad component of NMR resonance or the appearance of the narrow component on the background of broad component [49]. Such a change in the BL ${}^{1}$H NMR spectrum measured at 90 °C was observed for all the studied samples. Each spectrum is a superposition of one broad and one narrow component. The presence of a narrow component in the spectra clearly indicates the presence of polymer chains with higher mobility in the samples.

Moreover, narrowing and increasing of the amplitude of the narrow component in the BL ${}^{1}$H NMR spectra is markedly more visible for mixtures with nanoparticles than for pure PEI. This effect is more pronounced for higher concentration of nanoparticles and indicates the enhanced mobility of polymer chains in PEI after mixing with nanoparticles. The presence of ZnO and Al${}_{2}$O${}_{3}$ nanoparticles in PEI probably lead to the formation of loose structures [50] since nanoparticles can create steric hindrances for the proper cross-linking of the polymer network, which results in higher polymer chains mobility. The changes detected in the BL ${}^{1}$H NMR spectra are in accordance with DMA measurements, in which the shift of $T_{g}$ to lower values was observed for PEI mixed with nanoparticles. A similar lowering of $T_{g}$ values was found for SiO${}_{2}$/ epoxy [51] NCs, epoxy/graphene oxide vitrimer NCs [52] and epoxy/silica thermosets with vitrimers [53].

### 3.2. DMA Analysis of PEI Composites

From DMA measurements, the temperature associated with the peak magnitude of the mechanical loss factor (i.e., tan $\delta_{max}$) is referred to as the glass-transition temperature ($T_{g}$) [54]. A plot of the loss tangent curves for the corresponding samples can be found in Figure 3. The glass-transition temperature and height at tan $\delta_{max}$ for pure PEI, NC + Al${}_{2}$O${}_{3}$ and PEI + ZnO are around 113 °C, 97/93 °C and 106/97 °C, respectively. The width and height of the tan $\delta_{max}$ curve provide information about the cross-link density, which makes it possible to conclude that in the PEI there were many more cross-links than in the NCs. Since the NC cross-linking is lower, the loss tangent curves and peaks at lower temperatures have shifted. This occurs because the NPs caused a decrease in the degree of cross-linking and have lower glass-transition temperatures.

### 3.3. Dielectric Response Characterisation of PEI Composites

Since nanoparticles (NPs) are highly active, an important change in the electrical properties of the NCs can also occur at low nano-filler concentrations. For better understanding what effect NPs have on the dielectric properties of the pure PEI, we first describe the frequency characteristics of its complex permittivity. The frequency dependence of its complex relative permittivity for the temperature range 20–120 °C is depicted in Figure 4. For this type of PEI the real component of the complex relative permittivity ($\varepsilon_{r}$) is almost frequency-independent from 10 Hz, but there is a temperature dependence. With an increase of temperature to 80 °C it rises to 4.2 (50 Hz), and then with temperature it decreases to 2 (50 Hz) (Figure 4a). In the low-frequency region an increase of $\varepsilon_{r}$ is observed for a temperature of 80 °C and higher. This increase is caused by the influence of DC conductivity and electrode polarization [55]. Figure 4b shows the frequency dependence of the imaginary component of the complex relative permittivity ($\varepsilon_{i}$) on temperature that can be used to determine the polarization processes individually. At 40 °C we observe two interesting effects: an increase in $\varepsilon_{i}$ at sub-Hertz frequencies with a local maximum around 10 mHz (β-relaxation) and a slow increase at frequencies above 10 kHz.

The local maximum is characterized by the eigenfrequency $f_{e1}$ ($f_{e}=1/\left(2\pi \tau_{0}\right),\tau_{0}$—a characteristic time of relaxation process). This low-frequency maximum at 60 °C moves to 100 mHz and for higher temperatures it disappears. There is no visible high-frequency maximum, which can be attributed to the IDE relaxation (internal dipolar effect of the polymer chains) [39,56,57,58]. The eigenfrequency of this polarization process could be higher than 2 MHz. At frequencies below 10 mHz, there is a rapid increase $\varepsilon_{i}$ which is caused by DC conductivity together with electrode polarization. The second low-frequency local maximum caused by α-relaxation [39,42,58] at 12 mHz was observed at temperature 100 °C and it moves with temperature to 200 mHz (Table 3). In response to rising temperatures, chains become more mobile, what caused a decrease of their relaxation times and an increase of their eigenfrequency.

The next type of measurement was focused on the influence of various nanoparticles on the complex relative permittivity of pure PEI. Figure 5 shows its frequency dependence for PEI with 1 wt.% ZnO filler as a function of temperature. The significant increase in $\varepsilon_{r}$ at low frequencies for 60 to 100 °C is the main effect which can be seen as the first (Figure 4a). In the case of pure PEI (Figure 4a), $\varepsilon_{r}$ at all frequencies was less than 6. This increase is caused by interfacial polarization (IP) around the nanoparticles [38,39,40]. The frequency-independent area of $\varepsilon_{r}$ at 20 and 40 °C is in the whole frequency range and for higher temperatures for frequency above 10 Hz with value around 3.5. In the case of 120 °C, it has the value of only 2.5 for frequencies over 100 Hz. The $\varepsilon_{i}$ has no low-frequency local maximum at temperatures below 60 °C, but it slowly increases due to IDE—relaxation for frequencies above 10 kHz (Figure 4b). The effect of the α-relaxation process is highlighted by IP; it is observed from 80 °C (10 mHz) and it moves with temperature to higher frequencies. For 0.5 wt.% ZnO similar characteristics as for a higher concentration (Figure 5) were observed; the comparison is shown in Figure 6. A similar influence of ZnO on the dielectric properties of epoxy resin was presented in our previous works [32,59].

A comparison of the complex relative permittivity at temperatures of 60 and 100 °C in a wide frequency range for pure PEI and its composites with ZnO are shown in Figure 6. At first, it can be seen that NCs with 0.5 and 1.0 wt.% ZnO nanoparticles have a higher $\varepsilon_{r}$ than pure PEI (Figure 6a) for all frequencies. The imaginary permittivity for a 1% concentration of ZnO has higher values than for pure PEI. In the case of lower concentration $\varepsilon_{i}$ is lower at 60 °C and higher for frequencies from 100 mHz at 100 °C. The second important factor is the shift of the second local maxima ($f_{e2}$) at low-frequency due to both concentrations of ZnO NPs to higher values than in PEI (Table 3). The last important factor is that there is no observed local maximum $f_{e1}$ at low frequencies in the nanocomposites. For the lowest frequencies, higher values are observed due to the DC conductivity and electrode polarization.

The influence of various concentrations of Al${}_{2}$O${}_{3}$ on the complex relative permittivity of PEI at temperatures of 60 and 100 °C in a wide frequency range are shown in Figure 7. In this case both concentrations of Al${}_{2}$O${}_{3}$ cause a decrease of $\varepsilon_{r}$ to pure PEI at 60 °C. At a higher temperature of 100 °C, 1.0 wt.% Al${}_{2}$O${}_{3}$ has a higher $\varepsilon_{r}$ in the whole frequency range. The imaginary permittivity for 1.0 wt.% Al${}_{2}$O${}_{3}$ has higher values and for 0.5 wt.% Al${}_{2}$O${}_{3}$ has lower values than for pure PEI for both temperatures. The next important factor is the shift of the second local maximum ($f_{e2}$) at low-frequency due to 1.0 wt.% Al${}_{2}$O${}_{3}$ to higher values than in PEI. In the case of a lower concentration, this relaxation is observed at a similar eigenfrequency as PEI. The important factor is that again there is no observed local maximum $f_{e1}$ at low frequencies in the nanocomposites. For the lowest frequencies, higher values of $\varepsilon_{r}$ are observed due to the DC conductivity and electrode polarization, too.

However, real dielectrics have several variations of dipole molecules in different configurations; their relaxation times have some distribution. Parameters that characterize PEI and NCs can be obtained from the fits of the complex permittivity. We used the Cole-Cole model, which was widely used by other authors [42,58,60]. The resulting relation for the complex permittivity according to the Cole-Cole distribution with two relaxation process has the form:(1)$$\varepsilon^{*}=\varepsilon_{\infty }-j\frac{\sigma_{DC}}{\varepsilon_{0}\omega }+\frac{\Delta \varepsilon_{1}}{1+{\left(j\omega \tau_{1}\right)}^{1-a_{1}}}+\frac{\Delta \varepsilon_{2}}{1+{\left(j\omega \tau_{2}\right)}^{1-a_{2}}}.$$
where $\varepsilon_{\infty }$ is the permittivity at high-frequency, $\sigma_{DC}$ is the DC conductivity, ω=2πf, $\varepsilon_{0}$ is the permittivity of vacuum, Δε is the change of the permittivity at low- and high-frequency, τ is the relaxation time and *a* is distribution shape parameter, the subscripts 1 and 2 represent the β- or α-relaxation mode, respectively. The calculated parameters for the studied NCs are listed in Table 3. The bond and loose layers of the NPs caused an extension of the second peak, which corresponds to an increase of the shape parameter.

In previous sections how nanoparticles influence the complex relative permittivity were explained. It was shown that ZnO nanoparticles increase permittivity and for Al${}_{2}$O${}_{3}$ this was dependent on the concentration. In pure PEI and NCs local maximum’s were observed, which are characterized by the eigenfrequency. Their values and other parameters from the Cole-Cole model (Equation (Equation 1)) were obtained, see Table 3. Their next effect was a shift of the second local maximum ($f_{e2}$) of $\varepsilon_{i}$ to higher frequencies (Table 3). The conductivity for NC with ZnO was higher and for NC with Al${}_{2}$O${}_{3}$ was smaller than in pure PEI.

## 4. Discussion

Our measurements show several results. The addition of NPs to a commercial PEI caused important change in the relative permittivity at temperatures over 60 ${}^{\circ }$C for frequencies below 10 Hz. Further, the relative permittivity of NCs (PEI + ZnO) has higher values than pure PEI at all temperatures and frequencies (Figure 6a). The increase of relative permittivity can be attributed to the decrease of cross-linking in PEI matrix due to presence of nanoparticles [61]. This could be due better interaction of polymer chains with the nanoparticles, which in turn leads to higher mobility of polymer chains than in the PEI matrix. ZnO nanoparticles have the relative permittivity of 8.8 [62], which is higher than that of PEI, which means that NCs have a higher relative permittivity.

The value of relative permittivity is also influenced by the mobility of polymer chains, whose value rises as mobility increases. After ZnO or Al${}_{2}$O${}_{3}$ were inserted into the PEI matrix, the phase separation was disrupted. Compared with pure PEI, less hard phases are formed in nanocomposites. Through hydrogen bonding between ZnO nanoparticles and PEI, ZnO nanoparticles increased the inter-chain association. The reaction between the PEI and the hydroxyl groups on NPs is the cause of phase disruption in the PEI matrix [54,63,64]. A decrease in the hard phases is related to the higher mobility of polymer chains in composites, which also affects permittivity. ZnO NCs showed a higher increase in relative permittivity at temperatures above 100 ${}^{\circ }$C. This temperature is above $T_{g}$, the temperature at which polymer chains become more mobile. Based on the BL ${}^{1}$H NMR spectra (Figure 2), the increased chain mobility in NCs was confirmed, which also means that the chains are more connected to the NPs. The smaller widths of the both broad and narrow lines observed in both NCs suggest that the mobility of polymer chains in these samples is higher than that in the PEI. Incorporation of nanoparticles into PEI results in the formation of loose structure [65], since NPs can prevent the proper cross-linking of the polymer network. This can explain the higher polymer chains mobility in both mixtures. Similar changes in polymer mobility reflected in lower $T_{g}$ values were found for epoxy/silica thermosets with vitrimers [53]. The confirmation of chain mobility by two types of measurements (dielectric spectroscopy and NMR spectral analyses) is unique and has never been published before.

In the case of Al${}_{2}$O${}_{3}$ a decrease of the relative permittivity was measured, which can also be explained by the multi-core model from Tanaka [38,66]. The reason for the observed decrease at temperature 60 ${}^{\circ }$C is the result of the lower mobility of the polymer chains in the interfacial (bounded and bound layers) regions. The surface area of Al${}_{2}$O${}_{3}$ NPs is three times larger than ZnO, so there are more interchain associations through surface hydroxyl groups and covalent bonds [48,67] on NPs. Due to the increased connections between polymer chains and Al${}_{2}$O${}_{3}$, their mobility is higher than in the PEI matrix, thus the degree of cross-linking is lower. The surface of Al${}_{2}$O${}_{3}$ caused better connection with polymer chains, as with ZnO, so they are better bounded in the bound layer around the NPs. At the higher temperature of 100 ${}^{\circ }$C the situation was different due to the presence of more mobile polymer chains. In case of 1 wt.% $\varepsilon_{r}$ is same or higher as in pure PEI. In the case of a lower concentration, the chain mobility is not so high and due to other processes influencing the value of relative permittivity, its final value is smaller than pure PEI.

The difference in the conductivity and permittivity of the constituents of nanocomposites causes space-charge to accumulate at the interfaces of the NPs and the PEI matrix (bounded and bound layers). These tapped charges (electron and ion) [68] produce strong local electric field around the NPs, and the interfacial polarization is enhanced by the charge multiplication. Because it is higher than the Laplacian and geometric electric fields, it influences the reorienting of the electric dipoles of polymer chains bound around the NP. The higher electric field causes a faster transfer of the dipole charges, which is connected with the shift of the local maximum caused by the α-relaxation process to higher frequencies, and $\varepsilon_{r}$ increases more markedly at low frequencies.

Nanoparticles caused a more pronounced shift of the eigenfrequency of the α-relaxation process (the second local maximum) to higher frequencies (Figure 6b and Figure 7b), except for PEI + 0.5 wt.% Al${}_{2}$O${}_{3}$ (Table 3, Figure 7b). The α-relaxation becomes evident via the loss peaks in the $\varepsilon_{i}$ of the NCs spectra. The low-frequency region of these spectra shows α-relaxation in the form of a slight peak, which corresponds to the glass rubber transition of the PEI matrix. In the case of pure PEI, this can be observed from a temperature of 100 ${}^{\circ }$C at the eigenfrequency 11 Hz. Also, this relaxation was observed for other studied NCs, but at higher frequencies (Table 3, $f_{e2}$), than for pure PEI. This peak shifts to higher frequencies with increasing temperature, while the magnitude of the formed peaks decreases. On the other hand, the shift of eigenfrequency of α-relaxation to higher values or a decrease of α-relaxation time ($\tau_{2}$) is connected with increased filler content. This indicate that the decrease of the glass-transition temperature (Figure 3) shifts of $\tau_{2}$ to lower values is a function of filler content [58].

## 5. Conclusions

PEI nanocomposites with various concentrations of ZnO or Al${}_{2}$O${}_{3}$ nanoparticles were studied by dielectric frequency spectroscopy and NMR spectral analysis. For a description of the observed changes in the dielectric properties of the NCs, the multi-core model of NCs, the α-relaxation process and the influence of a local electric field on the trapped charges were used. In the case of ZnO filler, the complex permittivity had higher values than pure PEI. The increase of the relative permittivity was caused by the presence of highly mobile polymer chains in the interfacial regions around the NPs and the high permittivity of ZnO, Al${}_{2}$O${}_{3}$. The increase in the mobility of the polymer chains in the NCs was also confirmed by the NMR spectrum, which was connected with a decrease of cross-linking in matrix. The interconnection of these types of measurements to confirm the increasing mobility of the polymer chain is unique and has never been published before. DMA analysis showed that nanoparticles reduced the $T_{g}$ of NCs relative to pure PEI.

## Figures and Tables

**Figure 1 polymers-14-02202-f001:**
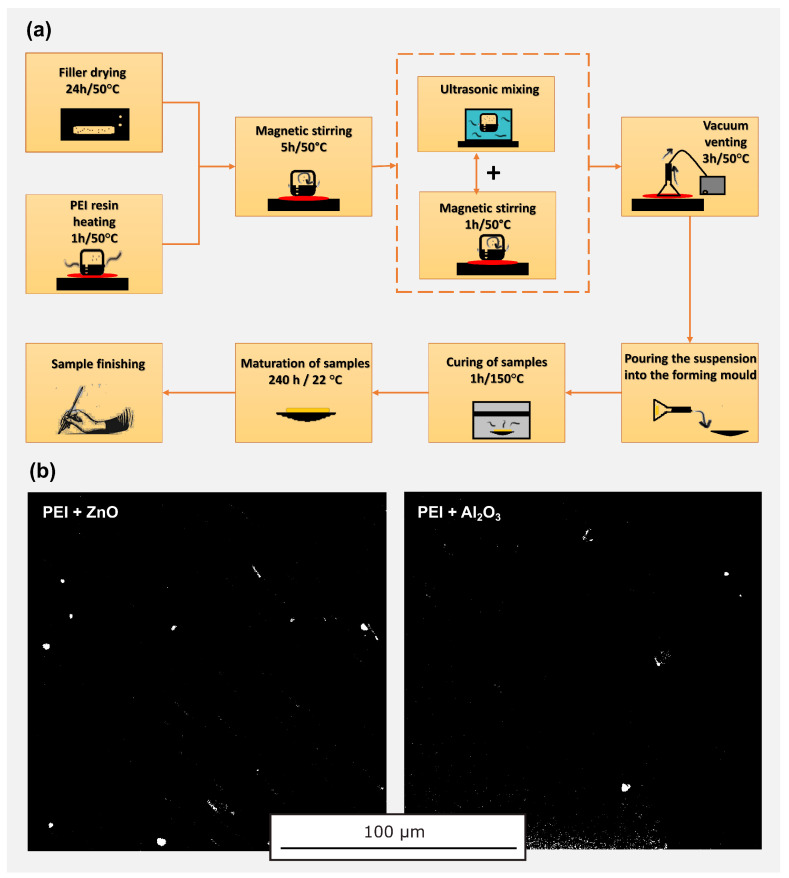
Preparation of PEI nanocomposites: (**a**) production diagram; (**b**) verification of particles dispersion by SEM.

**Figure 2 polymers-14-02202-f002:**
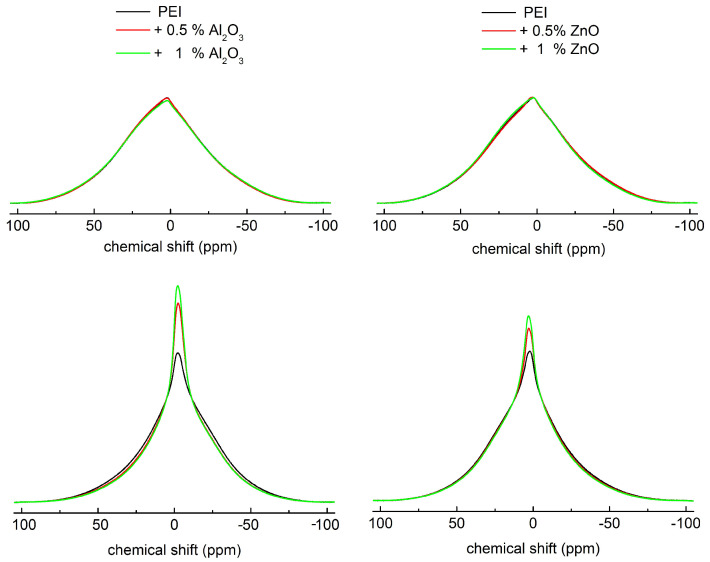
The BL ${}^{1}$H NMR spectra of pure PEI and its mixtures with Al${}_{2}$O${}_{3}$ or ZnO nanoparticles measured at ambient temperature (**upper row**) and 90 ${}^{\circ }$C (**lower row**).

**Figure 3 polymers-14-02202-f003:**
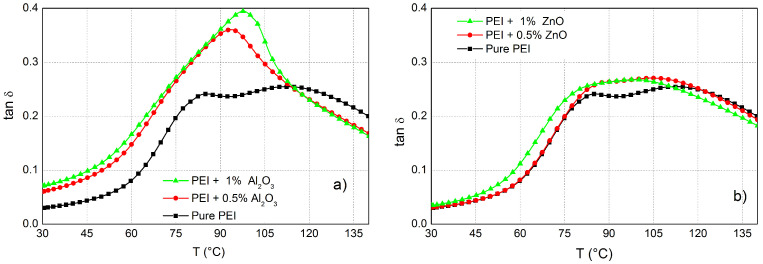
Temperature dependence of DMA development of pure PEI and its mixtures with Al${}_{2}$O${}_{3}$ (**a**) and ZnO (**b**) nanoparticles.

**Figure 4 polymers-14-02202-f004:**
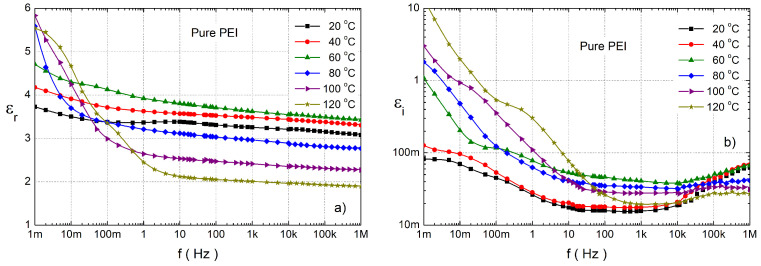
The frequency dependence of the relative real (**a**) and the imaginary (**b**) part of the complex relative permittivity for pure PEI at various temperatures.

**Figure 5 polymers-14-02202-f005:**
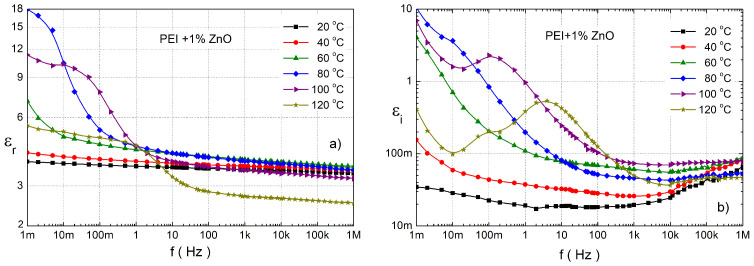
The frequency dependence of the real (**a**) and the imaginary (**b**) component of the complex relative permittivity for PEI with 1 wt.% ZnO nanoparticles.

**Figure 6 polymers-14-02202-f006:**
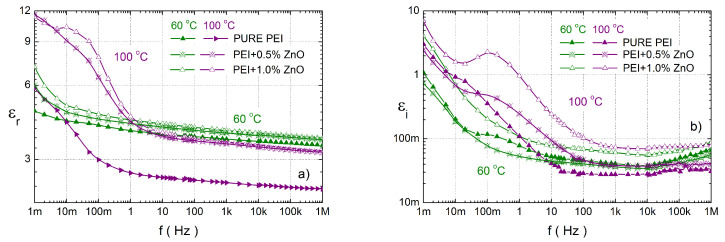
The frequency dependence of the real (**a**) and the imaginary (**b**) components of the complex relative permittivity for PEI and their various nanocomposites with ZnO at temperatures of 60 and 100 ${}^{\circ }$C.

**Figure 7 polymers-14-02202-f007:**
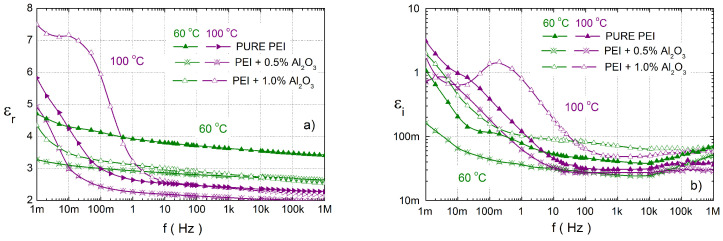
The frequency dependence of the real (**a**) aand imaginary (**b**) apart of the complex relative permittivity for PEI and their various nanocomposites with Al${}_{2}$O${}_{3}$ at temperatures of 60 and 100 ${}^{\circ }$C.

**Table 1 polymers-14-02202-t001:** The basic properties of used PEI resin.

Property	Value
Shelf life (23 °C)	6 months
Viscosity at 23 °C (mPa·s)	7500
Density at 23 °C (kg/m${}^{3}$)	1186
Gel time at 120 °C (min)	10 ± 2
Curing time at 150 °C (min)	60
Water absorption (following ISO 62)	
at 23 °C/24 h (mg)	3.4

**Table 2 polymers-14-02202-t002:** Technical parameters of used nanoparticles.

Parameter	ZnO	Al${}_{2}$O${}_{3}$
Diameter (nm)	20	20
Purity (%)	99+	99.97
Specific surface area (m${}^{2}$/g)	≥40	180
Bulk density (g/cm${}^{3}$)	0.1–0.2	3.95
Morphology of particles	spherical	spherical

**Table 3 polymers-14-02202-t003:** Parameters obtained from the fit by the Cole-Cole model for pure PEI and its various types of nanoparticles at 100 ${}^{\circ }$C ($\varepsilon_{r}$ and $\varepsilon_{i}$ at frequency 50 Hz, $\sigma_{DC}$ is the DC conductivity (10${}^{-12}$ S/m), τ is the relaxation time of α-relaxation, a is the shape parameter).

Parameter	$\varepsilon_{r}$ (60/100 ${}^{\circ }$C)	$\varepsilon_{i}$ (60/100 ${}^{\circ }$C)	$\sigma_{\mathrm{DC}}$	$\tau_{2}$	$f_{2e}$	$a_{2}$
Unit	-	-	(10${}^{-12}$ S/m)	(s)	(mHz)	-
Pure PEI	3.73 / 2.5	0.047/ 0.033	0.19	11.91	13	0.28
PEI + 0.5% ZnO	3.97/3.59	0.066/0.079	0.25	1.45	109	0.22
PEI + 1% ZnO	4.08/3.69	0.071/0.127	0.37	1.11	145	0.14
PEI + 0.5% Al${}_{2}$O${}_{3}$	2.15/2.82	0.029	0.13	13	12	0
PEI + 1% Al${}_{2}$O${}_{3}$	2.93/2.5	0.077	0.15	0.76	200	0.19

## Data Availability

The raw/processed data required to reproduce these findings cannot be shared at this time due to technical or time limitations.

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
