# Peer review of "Influence of Nanoparticles on the Dielectric Response of a Single Component Resin Based on Polyesterimide"

_polymers, 2022, doi:10.3390/polym14112202_

Round 1

Reviewer 1 Report

This manuscript presents a simple method to blend Al2O3 and ZnO nanoparticle fillers into PEI matrix and investigated its subsequent changes in complex relative permittivity. However, I believe that this manuscript still has the following problems and cannot meet our requirements.

  1. There are many problems with the English expressions in the article.

For example, “described in in relation” (Page 2 Line 40) should be changed as “described in relation”.

Page 3 Line 80, “by using by magnetic stirrer” should be changed as “by using magnetic stirrer”.

The authors should check the whole manuscript to avoid the problems.

  1. It’s said that the diameter of the nanoparticle fillers is 20 nm. However, in Table 2, the radius of the nanoparticle fillers is 20 nm. Authors should specify the size of the filler. Meanwhile, the authors should provide SEM pictures of the nanoparticle fillers.
  2. To confirm the dispersion of fillers in the matrix, the authors should provide electron micrographs of the composites.
  3. The sample preparation process described in the word is inconsistent with the process in Figure 1. For example, Page 3 Line 84, “an ultrasonic needle (20 kHz) simultaneously with a magnetic stirrer 84 (300 rpm) was applied for 1 h”. However, in Figure 1 it is stated as " ultrasonic mixing for 3h". Authors should further confirm the sample preparation process.
  4. The fabrication process of the composite is not clear.
  5. The manuscript uses DMA results, but does not provide specific data. Authors should provide DMA data.
  6. In Figure 3b, the result at 20 oC is lacking.
  7. Why choose 60 oC, 100 oC when studying the effect of different fillers on the complex permittivity of materials?
  8. Page 4 Line 117, “Figure A” should be changed as “Figure 2”.
  9. The image sequence number in the figures should be more obvious.
  10. There are also some problems in the references. For example, References 29 has unreadable code. Please check the references format.
  11. The English of this manuscript should be further polished. The authors should review the manuscript carefully to improve readability.

Author Response

Good afternoon.

Dear Reviewer.

Thank you for all your comments and suggestions, which have greatly contributed to the quality of the submitted article. Each your comments have been incorporated , as well as those of other reviewers, into the article, with some comments being the same from multiple reviewers. In the attached is a summary of our responses, which is simultaneously incorporated into the article.

Reviewer 2 Report

The paper shows results about impact of nanoparticles on dielectric response of component resin based on polyesterimid. Authors used dielectric frequency spectroscopy in order to measure the relationship between the real and imaginary parts of complex relative permittivity. The presence of weight concentration of nanoparticles in the PEI resin had influence on the segmental dynamics of the molecular chain and changed a charge distribution in the given system.

Dear authors. Thank you very much for your value paper about influence of nanoparticles on dielectric response of component resin based on polyesterimid. I have some comments and suggestions, what should be considered in presented to review paper.

Comments and suggestions:

  1. Introduction chapter well describes polymers materials, especially PEI, properties. The chapter also explain the impact of nano particles, especially Al2O3 and ZnO, on the increase or decrease of some materials properties, such as mechanical and electrical.
  2. Authors do not use paragraph in some parts of the paper – line 59, 70. Please correct.
  3. Fig.3.b, usually imaginary epsilon is compared to tan(delta) – dielectric losses. How do authors explain the decrease of this parameter with the increase of temperature? Usually, the increase of temperature means the increase of imaginary epsilon.
  4. Fig.6. Mostly, obtained results say, that used nano particles caused the increase of epsilon prim and bis. Is it good or not from practical point of view in authors opinion? Please explain if possible.

Author Response

(The authors gave the same response as above.)

Reviewer 3 Report

In this manuscript, the authors investigated the influence of various types of nanoparticles on the dielectric response of PEI matrix. I would suggest the acceptance of the manuscript after the following revision:

  1. The introduction section needs to be revised. (1) I suggest the authors to add a paragraph of literature review summarizing some commonly used nanoparticles in PEI matrix and their fabrication methods. (2) In addition, the novelty and advance of this work should be highlighted and illustrated. (3) some papers can be cited in the first paragraph regarding the use of polymer nanocomposites for different applications: Core/Shell Conjugated Polymer/Quantum Dot Composite Nanofibers through Orthogonal Non-Covalent Interactions." Polymers 8.12 (2016): 408.; Structural and mechanical properties of polymer nanocomposites." Materials Science and Engineering: R: Reports 53.3-4 (2006): 73-197.; "Device applications of polymer-nanocomposites." Biopolymers· PVA hydrogels, anionic polymerisation nanocomposites (2000): 163-205.
  2. Can authors comment on why the specific concentrations (0.5 and 1 wt%) of nanoparticles were chosen in the current study?
  3. I think the distribution uniformity of nanoparticles within the polymer matrix can impact the properties of the composites. I suggest the authors to conduct morphological analysis with TEM or SEM to study how nanoparticles were distributed within the polymer matrix. 
  4. Figure 5a, should the legend be "PEI + 1.0% ZnO" instead of "UP + 1.0% ZnO"?
  5. Lines 246-247 "Surface of Al2O3 caused better connection with molecular chains like with ZnO, so they are better bounded in the bound layer around the NPs." The binding mechanism between nanoparticles and polymers is not clearly explained. Are the surfaces of nanoparticles covered with ligands? Can the authors further clarify how the nanoparticles are connected with the polymer chains?
  6. Please check typos and grammar error over the entire manuscript.

Author Response

(The authors gave the same response as above.)

Round 2

Reviewer 1 Report

The manuscript has been revised, which can be accepted after revising the minor issue that the SEM image is too dark to see any information. 

Author Response

Dear Reviewer.
Thank you for your comment about SEM picture. This problen has been solved by enlarging the picture. The contrast of the background was improved to the higher quality.
Thank you for your positive review.